# Does hepatectomy improve outcomes of breast cancer with liver metastasis? A nationwide analysis of real-world data in Taiwan

Pin-Chun Chen[1,2]☯, Yuan-Chi Lee[2,3,4]☯, Yu-Chieh Su[4,5], Cheng-Hung Lee[6,7], Jian-Han Chen[3,4,8]‡*, Chung-Yen Chen[3,4,8]‡

**1** Division of Colon & Rectal Surgery, Department of Surgery, E-Da Hospital, Kaohsiung, Taiwan, **2** Division of General Surgery, E-Da Da-Chang Hospital, Kaohsiung, Taiwan, **3** Division of General Surgery, E-Da Hospital, Kaohsiung, Taiwan, **4** School of Medicine, College of Medicine, I-Shou University, Kaohsiung, Taiwan, **5** Division of Hematology-Oncology, E-Da Hospital, Kaohsiung, Taiwan, **6** Department of General Surgery, Dalin Tzu Chi Hospital, Buddhist Tzu Chi Medical Foundation, Chiayi, Taiwan, **7** School of Medicine, Tzu Chi University, Hualien, Taiwan, **8** Bariatric and Metabolism International Surgery Center, E-Da Hospital, Kaohsiung, Taiwan

☯ These authors contributed equally to this work.
‡ JH Chen and CY Chen are co-corresponding authors
* jamihan1981@gmail.com

**Data Availability Statement:** This study is based in part on data from the National Health Insurance Research Database (NHIRD), provided by the National Health Insurance Administration of the

## Abstract

### Background

Liver metastases from breast cancer are associated with poor prognosis, and treatment options are usually restricted to palliative systemic therapy. The impact of liver resection on metastasis remains controversial. The aim of this study is to investigate whether liver resection can offer better survival outcomes in cases of isolated liver metastases from breast cancer.

### Methods

We conducted a nationwide cohort study using a claims dataset from Taiwan's National Health Insurance Research Database (NHIRD). We identified all patients with breast cancer (diagnostic code ICD-9: 174.x) from the Registry for Catastrophic Illness Patient Database (RCIPD) of the NHIRD who underwent mastectomy between January 1, 2000, and December 31, 2008. Patients with other malignancies (history, initially, or during follow-up), those with a history of metastasis prior to or at initial admission for mastectomy, and those without liver metastases were excluded. Patients with other metastases between mastectomy and liver metastasis and those who died at first admission for liver resection were also excluded. All patients were followed up until December 31, 2013, or withdraw from the database because of death.

### Results

Data were analyzed for 1,116 patients who fulfilled the inclusion criteria (resection group: 89; non-resection group: 1,027). There were no differences in age, Charlson Comorbidity

Ministry of Health and Welfare and managed by the National Health Research Institutes (registration number NHIRD-103-246). The data utilized in this study cannot be made available in the manuscript, supplemental files, or in a public repository due to the "Personal Information Protection Act" executed by Taiwan's government, which took effect in 2012. Requests for data can be sent as a formal proposal to the NHIRD (http://nhird.nhri.org.tw) or via email to nhird@nhri.org.tw.

**Funding:** We are grateful to the grant support of E-Da Hospital, Taiwan, and IRB (EDAHI-109-001, EDAHI-110-003, EDAHP-106-060, EDAHP-111-007).

**Competing interests:** The authors have declared that no competing interests exist.

Index, or major coexisting diseases except renal disease between two groups. Kaplan–Meier analysis demonstrated that the liver resection group had significantly better overall survival (OS) than the non-resection group. (1-year: 96.6% vs. 52.3%, 2-year: 86.8% vs. 35.4%, 3-year: 72.3% vs. 25.2%, 5-year: 51.6% vs. 16.9%, respectively, p<0.001). Cox analysis revealed that the liver resection group exhibited a significant improvement in patient survival (hazard ratio [HR] = 0.321, 95% confidence interval [CI]: 0.234–0.440, p<0.001).

## Conclusion

These findings indicate that liver resection may offer better survival benefit in patients with breast cancer who develop new liver metastases post mastectomy.

## Introduction

Breast cancer is the second leading cause of cancer-related death in women worldwide [1] and the number of cases is increasing annually. Metastasis is present in many cases of breast cancer. Liver metastases (BCLM) develop in approximately 50% of all patients with metastatic breast cancer, representing the primary site of breast cancer recurrence in 5–12% [2]. BCLM is viewed as a disseminated disease, and the standard treatment focuses on systemic therapies and palliative local treatment [3]. In addition, metastatic breast cancer can lead to resistance to therapy and shorter overall survival. Thus, BCLM exhibit one of the worst prognoses among all types of breast cancer metastases [4]. Previous studies have reported that the median survival for BCLM is only 3–15 months, with a 5-year survival rate of only 0–12% [4–6]. BCLM has long been considered a systemic disease that requires only chemotherapy and optimal supportive care without surgical intervention [7,8]. With the advent of new chemotherapeutic agents and interventions other than hepatectomy (e.g., transarterial chemoembolization [TACE], hepatic arterial infusion chemotherapy [HAIC], and radiofrequency ablation [RFA]), hepatectomy appears to play a minor role in BCLM.

Surgical resection is a potentially curative treatment and is beneficial for overall survival patients with stage IV colorectal and neuroendocrine cancers who have developed liver metastases [9–11]. Previous studies have reported favorable outcomes in patients with breast cancer who underwent resection of brain [12,13] and bone metastases [14]. Hence, surgical resection may be a promising treatment option in cases of breast cancer with liver metastases (BCLM) [15]. Recent studies have demonstrated that liver resection can improve survival beyond 5 or 10 years after BCLM surgery [16–19].

There is no global consensus on whether liver resection is beneficial for patients with BCLM. The 5th European School of Oncology–European Society of Medical Oncology (ESO–ESMO) International Consensus Guidelines for Advanced Breast Cancer (ABC 5) suggest that liver resection for BCLM can be considered in select patients [20] who exhibit good performance status and have a limited tumor burden. However, the National Comprehensive Cancer Network (NCCN) guidelines do not mention liver resection as an option for BCLM [21]. Thus, the long-term survival benefit of hepatectomy for BCLM remains controversial.

In this cohort study, we extracted data from Taiwan's National Health Insurance Research Database (NHIRD) to assess the long-term effects of hepatectomy on survival in patients with BCLM. The null hypothesis is that hepatectomy did not offer survival benefit for BCLM compared to non-surgical treatment.

## Methods

### Database and study sample

The National Health Insurance (NHI) program was launched in Taiwan in 1995 and includes contracts with 97% of medical providers, covering approximately 23 million beneficiaries [22]. The National Health Insurance Research Database (NHIRD; registration number NHIRD-103-246) includes all claims data for beneficiaries and uses International Classification of Diseases, 9th Revision, Clinical Modification (ICD-9-CM) codes to record diagnoses [23]. This study was fully evaluated and approved by the Institutional Review Board of Da-lin Hospital (B10503009). All procedures performed in studies involving human participants followed the ethical standards of the institutional and national research committee and were in accordance with the 1964 Helsinki declaration and its later amendments or comparable ethical standards. The requirement for informed consent was waived given the nature of the study.

### Inclusion and exclusion criteria

We extracted data for all female patients with breast cancer (ICD-9 code: 174.x) diagnosed between January 1, 1996, and December 31, 2013, from the Registry for Catastrophic Illness Patient Database (RCIPD) of the NHIRD [24]. The RCIPD includes relatively accurate data regarding breast cancer diagnoses because pathological confirmation of breast cancer after surgery is required for patients to be registered. Based on inpatient expenditures (DD), we then identified all patients with diagnostic codes for breast cancer who had been admitted between January 1, 2000, and December 31, 2008, and had undergone mastectomy (partial mastectomy, ICD-9 procedure codes: 85.21 to 85.25; subcutaneous total mastectomy, ICD-9 procedure codes: 85.33 to 85.36; unilateral total mastectomy, ICD-9 procedure codes: 85.41, 85.43, 85.45, 85.47; and bilateral total mastectomy, ICD-9 procedure codes: 85.42, 85.44, 85.46, 85.48). We included patients who were admitted with the diagnostic code for liver metastasis (ICD-9 code: 197.7) after mastectomy for further analysis.

　　We excluded patients with metastasis before or at the time of mastectomy. We also excluded patients who had other malignancies or any metastasis before or during the follow-up period between mastectomy and the primary endpoint. Furthermore, patients who died during the first admission for liver metastasis were also excluded. All included patients were separated into two groups based on whether they underwent liver resection. The liver resection group included patients with ICD-9 procedure codes of 50.29 or 50.3, while the non-resection group included the remaining patients. The selection algorithm is illustrated in Fig 1.

### Index date and primary end points

The first admission date with liver metastasis was defined as the date of liver metastasis, which was regarded as the index date in this study. The primary endpoint was patient death, the date of which was identified using the RCIPD. If the date of death was not available in the RCIPD, death was defined as withdrawal from the NHI program [24]. All included patients were followed up until death or the end of the study period (December 31, 2013).

### Covariate assessment

Comorbidities (identified by ICD codes) were recorded in the NHIRD 1 year before the index date. Health status was assessed using the Charlson Comorbidity Index (CCI) [25,26]. Comorbidities used as covariates included diabetes mellitus (250), hypertension (401–405), liver

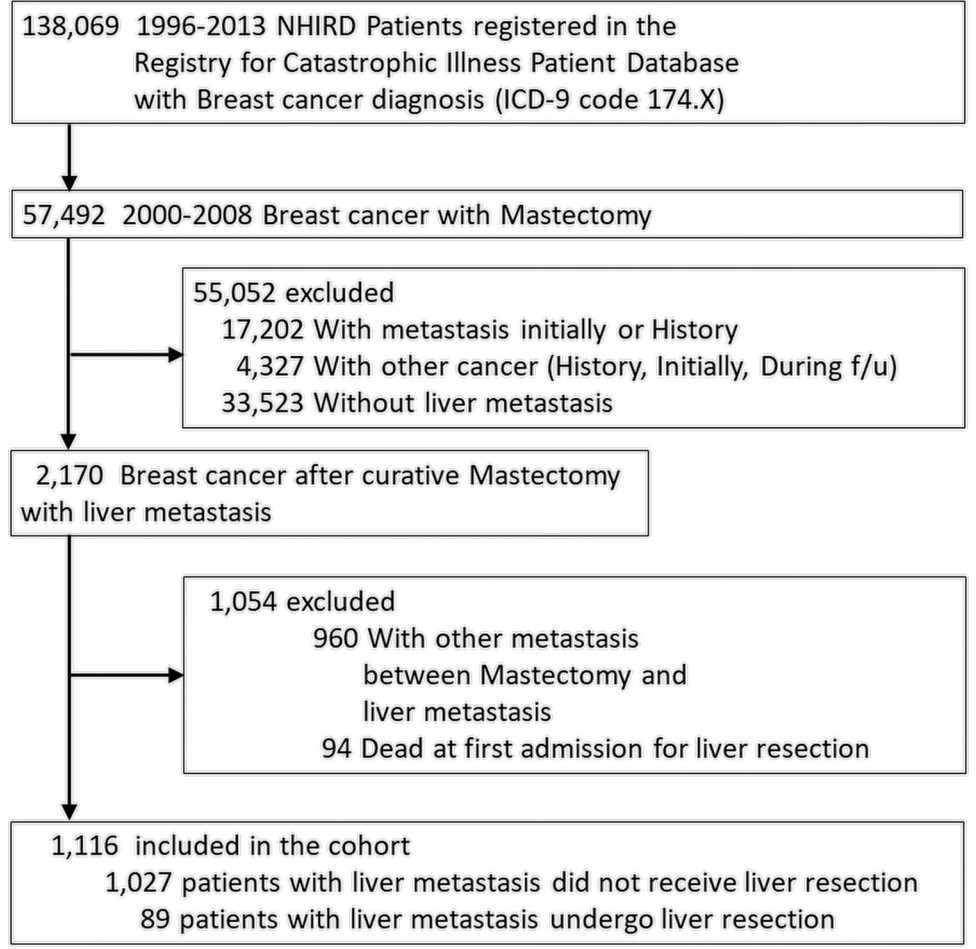

**Fig 1. Selection criteria.**

disease (571.2, 571.4–6, 572.2–8, 456.0–456.21), renal disease (582, 583, 585, 586, and 588), peptic ulcer disease (531–534), chronic pulmonary disease (490–496, 500–505, 506.4), cerebro-vascular disease (430–438), and myocardial infarction (410, 412).

## Statistical analysis

SPSS software (IBM, Chicago, IL, USA, Statistics 24 version) was used for the analysis. For basic clinical characteristics and covariates, the chi-square test and Fisher's exact test were used to compare categorical variables. Continuous variables were analyzed using the Kolmogorov–Smirnov test and then compared using Student's $t$-test or the Mann–Whitney U-test, based on the results of the Kolmogorov–Smirnov one-sample test. Kaplan–Meier analyses were used to compare differences in overall survival (OS) after liver metastasis between the liver resection and non-resection groups. We also used a univariable Cox proportional hazards model to evaluate the risk of overall mortality among the different covariates. Variables with a $p$ value less than 0.2 were selected and inserted into the multivariable backward stepwise Cox proportional hazards model. Statistical significance was defined as a two-tailed $p$ value $< 0.05$.

## Results

The analysis included 1,116 patients who developed liver metastases during the follow-up period after mastectomy. The median follow-up duration was 13.08 months. The liver resection and non-resection groups included 89 and 1,027 patients, respectively. The clinical characteristics, comorbidities, and follow-up durations of the two groups are presented in **Table 1**. Age, CCI values, and rates of major coexisting diseases (except renal disease) were similar between the two groups. The age at which mastectomy was performed, the frequency of total mastectomy, and the duration between mastectomy and liver metastasis were also similar in the two groups. The total follow-up duration was significantly longer in the liver resection group (40.30 ± 35.55 months) than in the non-resection group (10.77 ± 26.50 months). In the resection group, the median length of hospitalization was 9 days, and no mortality was observed.

### OS after more than 1 year of follow up

**Fig 2** demonstrates the results of the Kaplan–Meier analysis with log-rank testing for overall survival. The liver resection group exhibited significantly better OS after the identification of liver metastasis than the non-resection group. The 1-, 2-, 3-, and 5-year OS rates were 96.60%, 86.80%, 72.30%, and 51.60% in the liver resection group and 52.30%, 35.40%, 25.20%, and 16.90% in the non-resection group ($p<0.001$).

We included all covariates (age, CCI score, liver resection, and coexisting disease) in the Cox regression model (**Table 2**). Multivariate analysis revealed that liver resection had a

**Table 1. Basic characteristics in the liver resection and non-surgery groups.**

| Clinical characteristics | Non-resection group (N = 1,027) | Liver resection group (N = 89) | p |
|---|---|---|---|
| **Age at liver metastasis (y), median (IQR)** | 54.08 (16.42) | 53.75 (17.00) | 0.507 |
| **Charlson Comorbidity Index** | | | 0.685 |
| Median (IQR) | 2.00 (0) | 2.40 (0) | |
| Range | 0 (0–7) | 0 (0–4) | |
| **Follow up after liver metastasis (months)** | | | <0.001 |
| Median (IQR) | 10.77 (26.50) | 40.30 (35.55) | |
| **Major coexisting disease** | | | |
| DM | 103 (10.0%) | 10 (11.2%) | 0.714 |
| HTN | 176 (17.1%) | 17 (19.1%) | 0.661 |
| Liver disease | 5 (0.5%) | 0 | 1.000 |
| Chronic pulmonary disease | 30 (2.9%) | 1 (1.1%) | 0.506 |
| Cerebrovascular disease | 30 (2.9%) | 1 (1.1%) | 0.506 |
| Myocardial infarction | 6 (0.6%) | 1 (1.1%) | 0.442 |
| Peptic ulcer disease | 49 (4.8%) | 6 (6.7%) | 0.438 |
| Renal disease | 10 (1.9%) | 5 (5.6%) | 0.036 |
| **Age at Mastectomy (y), median (IQR)** | 50.67 (16.50) | 51.00 (16.0) | 0.448 |
| **Total Mastectomy** | 756 (73.6%) | 64 (71.9%) | 0.709 |
| **Time to liver metastasis (months)** | | | 0.409 |
| Median (IQR) | 34.10 (42.37) | 33.87 (35.90) | |
| **Time to liver resection** | | | |
| Median (IQR) | | 5.4 (12.72) | |
| Delayed | | 0.13–48.77 | |

IQR: Interquartile range; DM: Diabetes mellitus; HTN: Hypertension.

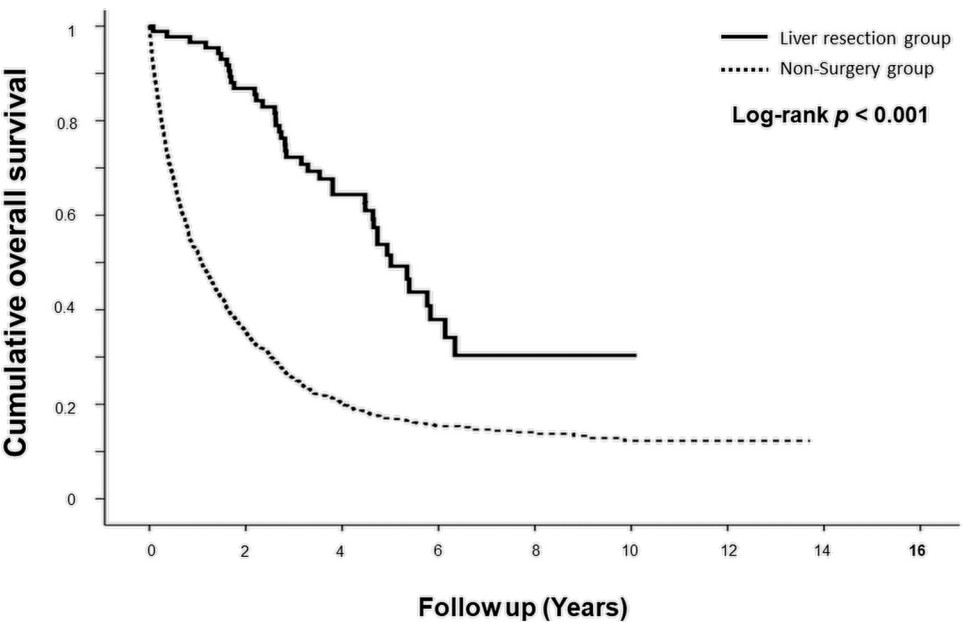

**Fig 2. Overall survival following identification of liver metastasis.**

significant survival benefit for patients with BCLM (hazard ratio [HR] = 0.308, 95% confidence interval [CI]: 0.224–0.424, $p<0.001$). Other risk factors for decreased overall survival included older age (HR = 1.008, 95% CI: 1.002–1.014, $p = 0.014$), history of hypertension (HR = 1.267, 95% CI: 1.038–1.545, $p = 0.020$), cerebrovascular disease (HR = 1.609, 95% CI: 1.085–2.385, $p = 0.018$), and myocardial infarction (HR = 3.561, 95% CI: 1.543–8.221, $p = 0.003$).

## Discussion

Our nationwide data analysis demonstrated that liver resection is beneficial for patients with BCLM, and that it is associated with relatively favorable long-term survival. The median 1-, 2-,

**Table 2. Risk factors influencing overall survival after identification of liver metastasis.**

| | Univariate analysis | | | | | | | Multivariate analysis | | | | | | | |
|---|---|---|---|---|---|---|---|---|---|---|---|---|---|---|---|
| | HR | | 95% CI | | | p | | HR | | 95% CI | | | p | | |
| **Age** | 1.013 | ( | 1.007 | - | 1.019 | ) | <0.001 | * | 1.008 | ( | 1.002 | - | 1.014 | ) | 0.014 | * |
| **CCI** | 1.141 | ( | 1.069 | - | 1.218 | ) | <0.001 | * | 1.013 | ( | 0.908 | - | 1.130 | ) | 0.816 | |
| **Liver resection** | 0.324 | ( | 0.237 | - | 0.444 | ) | <0.001 | * | **0.308** | **(** | **0.224** | **-** | **0.424** | **)** | **<0.001** | * |
| **Comorbidities** | | | | | | | | | | | | | | | |
| DM | 1.308 | ( | 1.048 | - | 1.631 | ) | 0.017 | * | 1.087 | ( | 0.857 | - | 1.379 | ) | 0.492 | |
| HTN | 1.457 | ( | 1.223 | - | 1.736 | ) | <0.001 | * | 1.267 | ( | 1.038 | - | 1.545 | ) | 0.020 | * |
| Liver disease | 2.427 | ( | 1.007 | - | 5.851 | ) | 0.048 | * | 1.748 | ( | 0.717 | - | 4.259 | ) | 0.219 | |
| Chronic pulmonary disease | 1.062 | ( | 0.702 | - | 1.607 | ) | 0.777 | | | | | | | | | |
| Cerebrovascular disease | 2.189 | ( | 1.508 | - | 3.176 | ) | <0.001 | * | 1.609 | ( | 1.085 | - | 2.385 | ) | 0.018 | * |
| Myocardial infarction | 3.000 | ( | 1.342 | - | 6.704 | ) | 0.007 | * | 3.561 | ( | 1.543 | - | 8.221 | ) | 0.003 | * |
| Peptic ulcer disease | 1.023 | ( | 0.742 | - | 1.411 | ) | 0.890 | | | | | | | | | |
| Renal disease | 1.148 | ( | 0.72 | - | 1.831 | ) | 0.563 | | | | | | | | | |

CCI: Charlson Comorbidity Index; CI: Confidence interval; HR: Hazard ratio; DM: diabetes mellitus; HTN: Hypertension.

3-, and 5-year OS rates were 96.60%, 86.80%, 72.30%, and 51.60% in the resection group and 52.30%, 35.40%, 25.20%, and 16.90% in the non-resection group, respectively ($p<0.001$). These finding were generally consistent with the available published data (Table 3). Furthermore, we identified several predictors of unfavorable OS in patients with BCLM undergoing hepatectomy, including myocardial infarction, older age, hypertension, and cerebrovascular disease.

**Table 3. Survival data of studies investigating surgical resection for BCLM patients.**

| Study, Year | Design | Survival rate, % | | Overall Survival (mo) (Non-surgery vs. Surgical resection) | Improved Survival Prognostic factors |
|---|---|---|---|---|---|
| Pocard et al, 2000 [27] | Retrospective, Single institution. n = 52 | 1-Year | 86% | 14 vs. 47 months | Interval between primary diagnosis and diagnosis of liver metastasis. (48 months) |
| | | 3-Year | 49% | | |
| | | 5-Year | NR | | |
| Elias et al, 2003 [28] | Prospective, Single institution. n = 54 | 1-Year | NR | NR vs. 34.3 months | Positive hormone receptor status |
| | | 3-Year | 50% | | |
| | | 5-Year | 34% | | |
| Adam et al, 2006 [5] | Prospective, Single institution. n = 85 | 1-Year | NR | NR | lack of response to prehepatectomy chemotherapy, The presence of extrahepatic metastases at the time of hepatectomy, R2 resection (negative association) |
| | | 3-Year | NR | | |
| | | 5-Year | 37% | | |
| Hoffmann et al, 2010 [29] | Prospective, Single institution. n = 41 | 1-Year | NR | NR vs. 58 months | Disease free interval less than 1 year, Positive resection margin (negative association) |
| | | 3-Year | 68% | | |
| | | 5-Year | 48% | | |
| Abbott et al, 2012 [30] | Prospective, Single institution. n = 86 | 1-Year | NR | NR vs. 57 months | Stable disease; positive estrogen receptor status |
| | | 3-Year | NR | | |
| | | 5-Year | 45% | | |
| Kostov et al, 2013 [31] | Prospective, Single institution. n = 42 | 1-Year | 84% | NR vs. 43 months | R0; diameter < 4 cm, Positive hormone receptor status |
| | | 3-Year | 64% | | |
| | | 5-Year | 38% | | |
| Mariani et al, 2013 [32] | Retrospective, Single institution. n = 51 | 1-Year | NR | NR vs. 91 months | Surgical resection; no extrahepatic disease |
| | | 3-Year | 80% | | |
| | | 5-Year | 50% | | |
| Bacalbasa et al, 2014 [33] | Prospective, Single institution. n = 43 | 1-Year | 93% | NR vs. 32.2 months | Positive hormone receptor status |
| | | 3-Year | 74% | | |
| | | 5-Year | 58% | | |
| Margonis et al, 2016 [34] | Retrospective, Muti-institution. n = 131 | 1-Year | 98% | NR vs. 53.4 months | Negative surgical margin, Diameter of BCLM (< 3cm) |
| | | 3-Year | 75% | | |
| | | 5-Year | NR | | |
| Sadot et al, 2016 [35] | Retrospective, Single institution. n = 69 | 1-Year | NR | 30 vs. 53 months | None |
| | | 3-Year | NR | | |
| | | 5-Year | 37% | | |
| Ercolani et al, 2018 [36] | Retrospective, Single institution. n = 51 | 1-Year | 92% | NR vs. 51 months | Tumor diameter (< 5cm), R0 resection, Triple-negative tumor (negative association) |
| | | 3-Year | 69% | | |
| | | 5-Year | 36% | | |
| Labgaa et al, 2018 [37] | Retrospective, Muti-institution. n = 59 | 1-Year | 92% | NR vs. 35 months | Age < 60 years |
| | | 3-Year | 74% | | |
| | | 5-Year | 61% | | |

*(Continued)*

**Table 3.** (Continued)

| Study, Year | Design | Survival rate, % | | Overall Survival (mo) (Non-surgery vs. Surgical resection) | Improved Survival Prognostic factors |
|---|---|---|---|---|---|
| Ruiz et al, 2018 [38] | Retrospective, Single institution. n = 139 | 1-Year | NR | 31 vs. 82 months | Not reported. |
| | | 3-Year | 81% | | |
| | | 5-Year | 69% | | |
| Sunden et al, 2020 [39] | Prospective, Muti-institution. n = 29 | 1-Year | 90% | 28 vs. 77 months | Surgical resection, HER2 gene amplification |
| | | 3-Year | 82% | | |
| | | 5-Year | 78% | | |
| He et al, 2020 [40] | Prospective, Muti-institution. n = 67 | 1-Year | 93% | NR vs. 57 months | Pringle maneuver, Increased interval between surgical resection and diagnosis of BCLM |
| | | 3-Year | 73% | | |
| | | 5-Year | 32% | | |
| Chun et al, 2020 [41] | Retrospective, Muti-institution. n = 136 | 1-Year | NR | 28 vs. 57 months | Breast cancer receptor status |
| | | 3-Year | NR | | |
| | | 5-Year | 45% | | |
| Ellis V. et al, 2021 [42] | Retrospective, Muti-institution. n = 98 | 1-Year | 91.1% | 28.8 vs. 55.2 months | Higher income status (income >$63,000), Insurance coverage, Surgical resection |
| | | 3-Year | 72.6% | | |
| | | 5-Year | 46.7% | | |
| Orlandi et al, 2021 [43] | Retrospective, Muti-institution. n = 22 | 1-Year | 100% | NR vs. 67 months | Negative resection margin (R0) |
| | | 3-Year | 85% | | |
| | | 5-Year | 65% | | |
| ProchAzkov, et al, 2021 [44] | Retrospective, Single institution. n = 30 | 1-Year | NR | NR vs. 56.3 months | Negative hormone receptor |
| | | 3-Year | 67% | | |
| | | 5-Year | 36% | | |

*NR: Not reported.

BCLM is viewed as a disseminated disease, which was reported with worst prognoses among all types of breast cancer metastases [4]. Previously, the standard treatment focuses on systemic therapies and palliative local treatment [3]. However, previous studies have reported that the median survival for BCLM is only 3–15 months, with a 5-year survival rate of only 8.5% [5,6].

Like our study, hepatectomy can offer a better survival for patients with BCLM, even beyond 5 or 10 years after BCLM surgery [16–19,45]. Ruiz A, et al. presented a case-matched analysis that liver resection for BCLM had an impressive median OS of 82 months when compared to a median OS of 31 months in BCLM patients who only received systemic treatment [38]. Compared to systemic treatment only, patients who underwent liver resection had significantly better mean (61.8 versus 38.6 months), 3-year(54.7% versus 45.6%), and 5-year OS (54.7% versus 21.9%, respectively) by using propensity score matching [46]. These findings are comparable to the results of the present study. In clinical practice, liver metastasectomy represents a possible therapeutic option for select patients with BCLM. In addition to being associated with better long-term outcomes, liver resection was found to be cost-effective in patients with BCLM when compared to systemic therapy alone, particularly in patients with ER-positive tumors or when newer targeted agents were used [47].

There are several factors influencing the survival after liver resection for BCLM. Tumor size also influences the survival rate among patients with BCLM. In a recent multi-institutional study, Margonis et al. analyzed data for 131 patients who underwent liver resection for BCLM between 1980 and 2014. They found that the median survival time for patients with tumors

<3.0 cm was 58.8 months, while that for patients with tumors ≥3.0 cm was 53.3 months (p = 0.041). Multivariate analysis indicated that a positive surgical margin (HR = 3.57, 95% CI: 1.40–9.16; *p* = 0.008) and a diameter greater than 3 cm (HR = 1.03, 95% CI: 1.01–1.06; *p* = 0.002) were associated with poorer survival [34]. Although we did not include the size of tumor and the extensive of liver resection for analysis because we cannot obtain these data from the database we used, all patients with BCLM who planned to received liver resection were well evaluated by surgeon. Patients in the resection group may have presented with a more acceptable oncologic burden for surgical resection.

Hormone receptor status is among the key factors considered when determining breast cancer treatment, especially in patients with BCLM [48]. Furthermore, molecular subtypes are not only a predictor of clinical outcomes in patients with BCLM; they are also a risk factor for liver metastasis [49,50]. Recently, a propensity-matched analysis of 136 patients who underwent hepatectomy plus systemic therapy reported that the intrinsic subtype was an independent predictor of poor OS (HR = 4.28) [51]. The median OS after resection among patients with luminal A, luminal B, HER2-enriched, and basal-like subtypes was 53, 75, 81, and 17 months (*p*<0.001), respectively.

Similarly, the median progression-free survival (PFS) among patients with the HER2-enriched subtype at 60 months was significantly better than that at 17, 16, and 5 months among patients with the luminal A, luminal B, and basal-like subtypes, respectively (*p*<0.001). After propensity score matching, the 5-year OS was significantly better in the surgical group than in the cohort of patients who had received systemic therapy alone (56% and 40%, *p* = 0.018). Lack of progesterone (PR) and estrogen receptor (ER) expression is associated with poor OS, as this reduces the response to hormonal therapy [52]. In recent years, however, there have been several important advances in targeting the unique biology of these subtypes, including several HER2 neu-targeted therapies. These subtypes unsurprisingly benefit the most from the resection of BCLM. Unfortunately, in our study, we cannot evaluate these reported finding such as status of hormone therapy and subtypes of tumor cell. The timing, regimen, and dosage of chemotherapy and hormone therapy were not recorded in the database and could not therefore be determined. However, in Taiwan, every patient who had breast cancer will receive chemotherapy, hormone therapy, anti-HER2 therapy and radiotherapies based on different condition. Most of these therapies were supported by national health insurance. Moreover, like previous reported study [16–19,38,45–47], the null hypothesis of our study is that hepatectomy did not offer survival benefit for BCLM compared to non-surgical treatment. Although we did not include these reported risk factor for analysis, we believed that even in patients with metachronous liver metastasis who received aggressive chemotherapy for metastatic gastric cancer, liver resection still had a role.

Recently, locoregional therapies have recently gained attention for their potential in the treatment of patients with BCLM [53]. With the advent of interventional treatments such as TACE, HAIC, and RFA, patients and clinicians have more treatment options. In a meta-analysis of 14 studies, Xiao et al. aimed to compare the therapeutic effectiveness of resection versus ablation among patients with BCLM. A comparison of patients who underwent RFA revealed that hepatic resection was associated with better 5-year OS (odds ratio [OR] = 0.38; 95% CI, 0.32–0.46; *p*<0.001) and 5-year disease-free survival (OR = 0.51; 95% CI, 0.40–0.66; *p*<0.001) [54]. Another meta-analysis of 23 studies revealed that hepatic resection resulted in longer median overall survival (mOS) and 5-year survival (45 months, 41%) than RFA (38 months and 11–33%) or TACE (mOS, 19.6 months; 1-year survival: 32–88.8%, n = 8 studies) [55]. In Taiwan, the most indication for RFA was hepatocellular carcinoma. On rare patients with metastatic liver tumor received RFA. Moreover, in the inclusion period of our study, before 2009, RFA for metastatic liver tumor is still debated. Although we did not exclude RFA from non-

resection group, the data we presented still reflects clinical significance in treating BCLM. However, these recent finding showed that a multidisciplinary approach and personalized treatment are important for managing patients with BCLM [48,56,57]. Further studies should aim to clarify which treatment provides the most benefit in patients with BCLM.

## Limitations

Our study had several limitations. First, some details including the actual initial stage, extension, and pathological characteristics of the primary tumor (e.g., ER, PR, or HER-2 status) and the details of each operation were not recorded and therefore could not be analyzed. Second, some details were not recorded in the database because of their nature, such as the timing, regimen, and dosage of chemotherapy. We were also unable to obtain details when patients received outpatient chemotherapy. Furthermore, in addition to selection bias, miscoding may have occurred since surgeons do not usually use ICD-9 coding but rather different coding and Health Insurance Surgical orders, which they obtain from the Taiwan NHI payment system. However, most ICD-9 codes during admission were assigned by professional coders based on records during admission. In addition, a code table comparing ICD-9 codes and NHI payment system codes is available from the National Health Insurance Administration Ministry of Health and Welfare. We therefore believe that the rate of miscoding for surgical procedures was limited. Moreover, the extent of tumor spread in the liver was not recorded in our database, indicating that patients in the resection group may have had a more acceptable oncologic burden on surgical resection. However, we believe that liver resection still played a role even in patients with metachronous liver metastasis who received aggressive chemotherapy for metastatic breast cancer.

## Conclusion

The present findings demonstrate that liver resection may offer survival benefit in patients with breast cancer who develop hepatic metastases post mastectomy. Based on our findings, liver resection could be considered as a treatment option for improving the overall survival of such patients. Further studies are required to validate the effectiveness of multidisciplinary approach and personalized treatment for BCLM.

## Acknowledgments

The authors would like to thank Dr. Yaw-Sen Chen, Division of General Surgery, E-Da Hospital, and Dr. Chao-Ming Hung, Division of General Surgery, E-Da Cancer Hospital for their assistance.

## Author Contributions

**Conceptualization:** Cheng-Hung Lee, Jian-Han Chen.

**Data curation:** Yu-Chieh Su, Jian-Han Chen.

**Formal analysis:** Cheng-Hung Lee, Jian-Han Chen, Chung-Yen Chen.

**Methodology:** Yu-Chieh Su, Jian-Han Chen, Chung-Yen Chen.

**Project administration:** Yu-Chieh Su, Jian-Han Chen, Chung-Yen Chen.

**Resources:** Cheng-Hung Lee, Jian-Han Chen.

**Software:** Jian-Han Chen.

**Validation:** Jian-Han Chen, Chung-Yen Chen.

**Visualization:** Yu-Chieh Su, Jian-Han Chen.

**Writing – original draft:** Pin-Chun Chen, Yuan-Chi Lee.

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
