## [Decision Letter · Decision Letter 0]

4 Jan 2022

PONE-D-21-34226Does hepatectomy improve outcomes of breast cancer with liver metastasis? A nationwide analysis of real-world data in TaiwanPLOS ONE

Dear Dr. Chen,

Thank you for submitting your manuscript to PLOS ONE. After careful consideration, we feel that it has merit but does not fully meet PLOS ONE’s publication criteria as it currently stands. Therefore, we invite you to submit a revised version of the manuscript that addresses the points raised during the review process.

We look forward to receiving your revised manuscript.

Kind regards,

Leonidas G Koniaris, MD

Academic Editor

PLOS ONE

Journal Requirements:

- https://journals.plos.org/plosone/article?id=10.1371%2Fjournal.pone.0182255

In your revision ensure you cite all your sources (including your own works), and quote or rephrase any duplicated text outside the methods section. Further consideration is dependent on these concerns being addressed.

Additional Editor Comments:

This is an interesting article but clearly overstates what may be concluded from a retrospective case series. Major revision is needed please specifically:

1) Clearly identify that this analysis supports the hypothesis that liver resection is beneficial for a subset of patients with isolated liver metastases from breast cancer.

2) Please put together a summary table of literature with the number of patients/power from other studies so that a more complete understanding of the current literature may be appreciated.

3) Please address the comments made by reviewer 1 as best as possible.

Reviewers' comments:

Reviewer's Responses to Questions

**Comments to the Author**

1. Is the manuscript technically sound, and do the data support the conclusions?

Reviewer #1: No

2. Has the statistical analysis been performed appropriately and rigorously? 

Reviewer #1: I Don't Know

3. Have the authors made all data underlying the findings in their manuscript fully available?

Reviewer #1: Yes

4. Is the manuscript presented in an intelligible fashion and written in standard English?

Reviewer #1: Yes

5. Review Comments to the Author

Reviewer #1: The authors address an important question of liver resection in BCLM. However, the study is lacking in strength. The authors acknowledge this deficiency but persist to draw major conclusion. The discussion merely is review of literature and doesn't discuss the study findings being validated by other similar studies. no data on adj therapies or on liver ablative therapies in non resection group. no data on extent of liver disease, type of resection, tumor biology, adjuvant or neoadj therapies. While the authors acknowledge the limitations, they still draw major conclusion that liver resection is beneficial. While this may be true, the data is lacking and insufficient to rule out confounding factors.

6. PLOS authors have the option to publish the peer review history of their article (what does this mean?). If published, this will include your full peer review and any attached files.

Reviewer #1: No

---

## [Author Response · Author response to Decision Letter 0]

3 Mar 2022

Q: Reviewer #1: The authors address an important question of liver resection in BCLM. However, the study is lacking in strength. The authors acknowledge this deficiency but persist to draw major conclusion. The discussion merely is review of literature and doesn't discuss the study findings being validated by other similar studies. no data on adj therapies or on liver ablative therapies in non resection group. no data on extent of liver disease, type of resection, tumor biology, adjuvant or neoadj therapies. While the authors acknowledge the limitations, they still draw major conclusion that liver resection is beneficial. 

While this may be true, the data is lacking and insufficient to rule out confounding factors.

A: 

Thank you for your valuable opinions. 

We have made the discussion between our study and those reviewed literates of the adjuvant therapy, and other therapies for treatment of liver mets from breast cancer, address the limitation in discussion section and marked. 

Also, we have adjusted our conclusion and marked. 

Conclusion in abstract:

These findings indicate that liver resection may offer better survival benefit in patients with breast cancer who develop new liver metastases post mastectomy.

Conclusion in Article:

The present findings demonstrate that liver resection may offer survival benefit in patients with breast cancer who develop hepatic metastases post mastectomy. Based on our findings, liver resection could be considered as a treatment option for improving the overall survival of such patients. Further studies are required to validate the effectiveness of multidisciplinary approach and personalized treatment for BCLM.

---

## [Decision Letter · Decision Letter 1]

31 Mar 2022

Does hepatectomy improve outcomes of breast cancer with liver metastasis? A nationwide analysis of real-world data in Taiwan

PONE-D-21-34226R1

Dear Dr. Chen,

We’re pleased to inform you that your manuscript has been judged scientifically suitable for publication and will be formally accepted for publication once it meets all outstanding technical requirements.

Kind regards,

Leonidas G Koniaris, MD

Academic Editor

PLOS ONE

Additional Editor Comments (optional):

Reviewers' comments:

Reviewer's Responses to Questions

**Comments to the Author**

1. If the authors have adequately addressed your comments raised in a previous round of review and you feel that this manuscript is now acceptable for publication, you may indicate that here to bypass the “Comments to the Author” section, enter your conflict of interest statement in the “Confidential to Editor” section, and submit your "Accept" recommendation.

Reviewer #1: All comments have been addressed

2. Is the manuscript technically sound, and do the data support the conclusions?

Reviewer #1: Yes

3. Has the statistical analysis been performed appropriately and rigorously? 

Reviewer #1: Yes

4. Have the authors made all data underlying the findings in their manuscript fully available?

Reviewer #1: Yes

5. Is the manuscript presented in an intelligible fashion and written in standard English?

Reviewer #1: Yes

6. Review Comments to the Author

Reviewer #1: (No Response)

7. PLOS authors have the option to publish the peer review history of their article (what does this mean?). If published, this will include your full peer review and any attached files.

Reviewer #1: **Yes: **Santosh Nagaraju

---

## [Editor Report · Acceptance letter]

13 Apr 2022

PONE-D-21-34226R1 

Does hepatectomy improve outcomes of breast cancer with liver metastasis? A nationwide analysis of real-world data in Taiwan 

Dear Dr. Chen:

I'm pleased to inform you that your manuscript has been deemed suitable for publication in PLOS ONE. Congratulations! Your manuscript is now with our production department. 

Kind regards, 

on behalf of

Dr. Leonidas G Koniaris 

Academic Editor

PLOS ONE